# NEUROD1 Is Required for the Early α and β Endocrine Differentiation in the Pancreas

**DOI:** 10.3390/ijms22136713

**Published:** 2021-06-23

**Authors:** Romana Bohuslavova, Ondrej Smolik, Jessica Malfatti, Zuzana Berkova, Zaneta Novakova, Frantisek Saudek, Gabriela Pavlinkova

**Affiliations:** 1Institute of Biotechnology CAS, 25250 Vestec, Czech Republic; romana.bohuslavova@ibt.cas.cz (R.B.); ondrej.smolik@ibt.cas.cz (O.S.); jess.malfatti@gmail.com (J.M.); zaneta.novakova@ibt.cas.cz (Z.N.); 2Department of Cell Biology, Faculty of Science, Charles University, 12843 Prague, Czech Republic; 3Laboratory of Pancreatic Islets, Institute for Clinical and Experimental Medicine, 14021 Prague, Czech Republic; zube@ikem.cz (Z.B.); frsa@ikem.cz (F.S.)

**Keywords:** transcriptional network, pancreatic development, mouse model, genetic mutation, NEUROD1

## Abstract

Diabetes is a metabolic disease that involves the death or dysfunction of the insulin-secreting β cells in the pancreas. Consequently, most diabetes research is aimed at understanding the molecular and cellular bases of pancreatic development, islet formation, β-cell survival, and insulin secretion. Complex interactions of signaling pathways and transcription factor networks regulate the specification, growth, and differentiation of cell types in the developing pancreas. Many of the same regulators continue to modulate gene expression and cell fate of the adult pancreas. The transcription factor NEUROD1 is essential for the maturation of β cells and the expansion of the pancreatic islet cell mass. Mutations of the *Neurod1* gene cause diabetes in humans and mice. However, the different aspects of the requirement of NEUROD1 for pancreas development are not fully understood. In this study, we investigated the role of NEUROD1 during the primary and secondary transitions of mouse pancreas development. We determined that the elimination of *Neurod1* impairs the expression of key transcription factors for α- and β-cell differentiation, β-cell proliferation, insulin production, and islets of Langerhans formation. These findings demonstrate that the *Neurod1* deletion altered the properties of α and β endocrine cells, resulting in severe neonatal diabetes, and thus, NEUROD1 is required for proper activation of the transcriptional network and differentiation of functional α and β cells.

## 1. Introduction

All forms of diabetes mellitus are characterized by the dysfunction and reduction of insulin-producing β-cells. The most critical step for understanding the pathophysiology of diabetes and for restoring lost and dysfunctional endocrine β cells is the identification of molecular cues that can be used for direct transformation in situ. Besides β cells, the pancreatic endocrine islets contain four additional hormone-secreting cell types; glucagon-secreting α cells, somatostatin-releasing ∂ cells, pancreatic polypeptide (PPY)-secreting PP cells, and a minority of ghrelin-producing ε cells [1,2]. In the mouse, all pancreatic endocrine cells differentiate from PDX1^+^ multipotent progenitors that transiently expressed *Neurogenin 3* (*Neurog3*) [3]. The first endocrine cells appearing in the dorsal pancreatic bud at embryonic day (E) 9.5 are cells expressing glucagon and ghrelin [4,5,6]. The insulin expressing cells are detected around E11.5, followed by somatostatin^+^ and PPY^+^ cells [4,5]. Endocrine cell formation coincides with the two morphogenesis stages, the primary (E9.0–E12.5) and secondary (E13.5–E15.5) epithelial-to-mesenchymal transitions, during mouse pancreas development [7]. The formation of the endocrine islets begins with the secondary transition, and immature differentiated α and β cells exponentially proliferate to produce the islet cell mass [7,8,9]. These functionally immature cells undergo maturation to obtain a hormone-producing glucose-responsive phenotype during the late fetal and postnatal period [9].

Complex interactions of transcription factor networks regulate the specification, differentiation, expansion, and maturation of endocrine cells in the developing pancreas (as reviewed in [2,10]. NEUROD1, a basic helix–loop–helix (bHLH) transcription factor, is crucial for pancreatic development, as mice with deletions of *Neurod1* die perinatally due to severe diabetes [11,12]. Mutations in the *NEUROD1* gene in humans are linked to maturity-onset diabetes of the young (MODY), specifically MODY6 [13,14], and to susceptibility to the acute-onset of type I diabetes mellitus [15]. Inactivation of *Neurod1* in human embryonic stem cells results in failure to activate a β cell transcriptional network and differentiate into functional β cells [12]. NEUROD1 together with PDX1, ISL1 and MAFA, are key transcription factors regulating insulin synthesis in pancreatic β cells in response to blood glucose [16]. Consistently with these observations, conditional deletion of *Neurod1* in the insulin-producing cell population at the onset of their formation during pancreas development results in severe glucose intolerance and immature β cell characteristics, although these mice formed islets comparable in size to controls and survived to adulthood [17].

During mouse pancreatic development, *Neurod1* mRNA is first expressed in the pancreatic primordium at E9.5 along with the first glucagon^+^ cells and continues to be expressed in the endocrine precursors of α and β cells during the first and second transition wave in the mouse [11]. Although glucagon-producing α and insulin-producing β cells are found during the secondary transition in the *Neurod1*-null pancreas, *Neurod1* elimination results in the failure to increase β-cell mass due to reduced proliferation at E17.5 [12] or apoptosis [11], as it is in dispute. This fact suggests a functional role of NEUROD1 during the expansion phase of β cells. However, the role of NEUROD1 in early pancreas development and the differentiation of α and β cells is still unclear.

Given the importance of NEUROD1 in diabetes research, we examined different aspects of NEUROD1 requirements for pancreas development in this study. We eliminated *Neurod1* during early pancreas development and performed detailed molecular analyses of primary and secondary transitions, and subsequent α and β cell differentiation, and organization of pancreatic islets. For the first time, we demonstrated the effects of NEUROD1 deficiency during the primary transition for the formation of the endocrine precursor population. Additionally, we showed that the elimination of *Neurod1* impairs the expression of key transcription factors regulating pancreatic endocrine cell differentiation during the secondary transition of pancreas development, and thus, negatively, affecting the formation of α and β cells. Comparative molecular analyses revealed that the deletion of *Neurod1* affected the transcriptional networks important for α- and β-cell differentiation and the proliferation potential of β cells, resulting in a significant reduction of functional islet cell mass, and disorganization of islet architecture in the developing pancreas.

## 2. Results

### 2.1. Conditional Deletion of Neurod1 Results in Neonatal Diabetes

To assess the role of *Neurod1* in early pancreatic development, we introduced a somatic *Neurod1* mutation using a floxed allele (*Neurod1^loxP/loxP^*) [18] and an *Isl1^Cre^* [19], generating *Neurod1CKO* conditional mutants with a *Neurod1^loxP/loxP^;Isl1^Cre/+^* genotype (breeding scheme in Figure 1A). We compared *Neurod1CKO* mutants to control animals of *Neurod1^loxP/loxP^* or *Neurod1^loxP/+^* genotypes. *Neurod1CKO* embryos were recovered at expected Mendelian ratios in the examined embryonic days (Figure 1B). However, *Neurod1CKO* did not survive postnatally due to severe neonatal diabetes (Figure 1C), corresponding to the phenotype of the global deletion of *Neurod1* [11].

### 2.2. Altered Formation of Endocrine Cells in Neurod1CKO during the Primary Transition of Pancreas Development Corresponds to NEUROD1 Elimination

Pancreas development begins with PDX1 expression and evagination of a dorsal pancreatic bud from the foregut endoderm around E8.5, followed by the formation of the ventral pancreatic bud [2]. PDX1 specifies multipotent pancreatic progenitors and is required for the formation of NEUROG3^+^ endocrine precursors [20]. In differentiating endocrine cells, PDX1 is restricted to β cells, and is downregulated in cells directed to the α fate, glucagon-expressing cells [21]. A loss of NEUROD1 expression and the formation of cell clusters co-expressing NEUROD1 and Isl1-Cre in *Neurod1CKO* was already detected in the PDX1^+^ pancreatic domain as early as E9.5 (Figure 2A). There were diminished delaminating NEUROD1^+^ clusters with a significant loss (61%) of NEUROD1 in the *Neurod1CKO* dorsal pancreas compared to the littermate controls at E10.5 (Figure 2A,B). The first sign of endocrine cell differentiation is the formation of glucagon-expressing cells in the dorsal pancreatic bud during the primary transition [5]; therefore, we first evaluated glucagon expression (Figure 3A,B). There was an increased number of cells co-expressing PDX1 and glucagon in the *Neurod1CKO* pancreas, indicating abnormalities in the differentiation of endocrine precursors (Figure 3A′,B′). The formation of glucagon^+^ cell clusters was unaffected in *Neurod1CKO* during the primary transition of the developing pancreas (Figure 3C–E). In contrast to littermate controls, insulin expressing cells were not detected in the *Neurod1CKO* pancreas at E12.5, indicating a delay in the differentiation of insulin^+^ endocrine cells (Figure 3F,G). Interestingly, there was a moderate reduction of cells expressing NEUROG3, a marker of early endocrine progenitors, in the dorsal pancreas of *Neurod1CKO* compared to littermate controls at E11.5 (Figure 3H–J), suggesting a possible regulatory feedback loop between *Neurod1* and *Neurog3* during the primary transition of pancreatic development.

### 2.3. Differentiation of α and β Cell Lineages Is Affected in the Neurod1CKO Pancreas during the Early Phase of the Secondary Transition

The period of the secondary transition is characterized by a major wave of endocrine cell differentiation, the formation of endocrine protoislets consisting of *Neurog3^+^* endocrine progenitors delaminating from the bipotent trunk epithelium, and the presence of all five hormone-expressing endocrine lineages [2]. To define changes in endocrine differentiation in the *Neurod1CKO* pancreas during the secondary transition, we quantified mRNA levels of genes encoding transcription factors necessary for the differentiation of α and β cell lineages (Figure 4A), and pancreatic endocrine hormones (Figure 4B). These data indicate that the elimination of *Neurod1* affected the formation of pancreatic endocrine α and β populations. We found a significant reduction in the mRNA expression of *Arx*, *Pou3f4*, *Pax6,* and *MafB* transcription factors that affect aspects of α-cell fate and differentiation [22,23,24,25]. *Pax6* and *MafB* transcription factors are also crucial for other pancreas endocrine cell differentiation, particularly β cells [22,26]. Significantly decreased expression was also detected for *MafA, Pax4, Insm1, Foxa2, Nkx2.2*, and *Pdx1* transcription factors that have been shown to regulate β cell differentiation [22,27,28,29,30]. Except for *MafB*, *Neurog3*, and *Pou3f4*, all these genes have been identified as direct targets of NEUROD1 in ChIP-Seq analysis of regulatory regions in murine islets [31]. Although all pancreatic endocrine cells differentiate from NEUROG3^+^ progenitors in mice, there are sequential competence states of these precursors to generate a subset of endocrine cell types [5,32]. Our qPCR analyses showed changes only in the expression of genes encoding insulin and glucagon endocrine hormones, but not PPY, ghrelin, and somatostatin (Figure 4B). Notably, the expression of *Neurog3* was reduced in *Neurod1CKO.* This trend corresponds to a decrease in the number of cells expressing NEUROG3 at E11.5 (Figure 3J), indicating a negative effect of *Neurod1* elimination on the formation of the endocrine NEUROG3^+^ progenitors. To determine whether decreased *Neurog3* levels in the *Neurod1CKO* pancreas were due to decreased proliferation of NEUROG3^+^ progenitors, we evaluated proliferation of NEUROG3^+^ cells in the developing pancreas at E15.5, at which there is a peak in *Neurog3* expression [5,32]. The numbers of proliferating NEUROG3^+^ cells were increased in the *Neurod1CKO* pancreas compared to controls (Figure 5).

### 2.4. Deletion of Neurod1 Is Associated with Reduced Insulin Production, Disorganized Architecture of Islets of Langerhans, and Reduced Proliferation of β Cells

Next, we evaluated the formation of islets of Langerhans and the differentiation of α and β cells. The secondary transition wave from E13.5 to E15.5 is characterized by a massive differentiation of pancreatic lineages from delaminating epithelial cells [1,2,9]. Starting at E16, endocrine α and β cells proliferate to increase islet cell mass [9]. Consistent with the previous study [11], we found glucagon-producing α and insulin-producing β cells in the *Neurod1CKO* pancreas (Figure 6). While β cells co-expressing insulin and PDX1 (a differentiation marker for β cells) were present at E15.5, insulin production appeared reduced in *Neurod1CKO* (Figure 6A–B′). The reduced insulin production corresponds to the quantification of mRNA levels of insulin genes, *Ins1* and *Ins2,* in the E14.5 *Neurod1CKO* pancreas. Compared to the control islets with β cells found in the core and α cells at the periphery, the *Neurod1CKO* architecture of islets of Langerhans was disorganized, with both cell types intermingled within the islets, as shown at E17.5 (Figure 6C–D′).

Then, we questioned whether the expansion of β-cells is affected in the *Neurod1CKO* pancreas at E17.5, during a period characterized by a major expansion of the β cell population and islet cell mass production. Previous studies using *Neurod1*-null mice [11] and mice with a conditional *Neurod1* deletion in the NEUROG3^+^ endocrine cells by *Neurog3^Cre^* [12] open to dispute the main reason for the diabetic phenotype and perinatal lethality of these *Neurod1* mutants. The studies concluded that β cells differentiate during the second transition but fail to form islets due to reduced proliferation at E17.5 [12] or apoptosis [11]. To resolve this contradiction, we compared proliferation and apoptosis in β cells in our *Neurod1CKO* mutant at E17.5 (Figure 7). Consistent with previous reports, the morphometric quantification indicated that the insulin-producing cell mass was significantly decreased at E17.5 (Figure 7G). At this time point, we found a significant reduction of proliferating β cells (Figure 7A,B,A′,B′,H) without noticeable increase in apoptotic cells in *Neurod1CKO*, as evaluated by TUNEL (Figure 7C,D, the Appendix A). We also analyzed apoptosis in the pancreas of newborn pups at P0, as at this stage, substantial apoptosis was also reported for *Neurod1-null* mice [11]. We did not detect measurably increased apoptosis in the *Neurod1CKO* pancreas (Figure 7E,F). This fact argues that the proliferative potential of β cells in the *Neurod1CKO* islets is significantly reduced, ruling out any significant contribution of apoptosis to the loss of β-cell mass.

### 2.5. Abnormalities in the Expression of Pancreatic Endocrine Markers Are Associated with a Severe Diabetic Phenotype of Neurod1CKO after Birth

Because of the abnormalities in the formation of islets of Langerhans at E17.5 with the altered distribution of α and β cells, and severe neonatal diabetes in *Neurod1CKO,* we further investigated the formation of islets and insulin production at P0. We found clusters of cells expressing PDX1, a marker of differentiated β cells, without any detectable insulin expression in *Neurod1CKO* (arrows in Figure 8B). Compared to α cells in the control pancreas (Figure 8A), we also found cells co-expressing glucagon and PDX1, indicating abnormalities in the differentiation of α cells (Figure 8B, arrowheads). Since insulin production was noticeably reduced in *Neurod1CKO*, we evaluated the protein level of proinsulin 1 and proinsulin 2, encoded by the *Ins1* and *Ins2* genes, respectively. Rodents express two nonallelic insulin genes, *Ins1* and *Ins2,* and double deficiency for *Ins1* and *Ins2* results in acute diabetes and neonatal death [33]. Interestingly, the single *Ins1^−/−^* or *Ins2^−/−^* deletion mutants survive by β-cell mass increase to compensate for low insulin production [34]. We used anti-C-peptide 1 or anti-C-peptide 2 antibodies equimolar to insulin 1 and insulin 2, respectively, to compare the expression levels in pancreatic sections between *Neurod1CKO* and controls. A dramatic decrease in immunostaining for C-peptide 1 compared to C-peptide 2 was detected in the remaining β cells in the P0 *Neurod1CKO* pancreas (Figure 8C–F). Diminished levels of C-peptide 1 were also found in the *Neurod1CKO* pancreas at E17.5 (Figure 8G,H), indicating a reduced activation of the *Ins1* gene during embryonic development. These data are consistent with the previous study showing that deletion of *Neurod1* results in a loss of *Ins1* expression, whereas *Ins2* expression is relatively unaffected [17]. We used RT-qPCR to quantify the expression of hormones and selected β cell markers (Figure 8I). Significantly decreased insulin and glucagon mRNA levels corresponded to the reduction of α and β cells noted in the immunohistochemical morphometric analyses of the *Neurod1CKO* pancreas. We also detected a significant decrease in the mRNA *Pdx1* [28] and *MafA* [35], essential markers of differentiated β cells, and reduced *Pax6* levels indicating decreased endocrine cells in the *Neurod1CKO* pancreas [36]. Thus, abnormalities in the cytoarchitecture of islets of Langerhans, a failure to generate β-cell mass, and deficiency in *Ins1* expression correlated with severe neonatal diabetes and postnatal lethal phenotype of *Neurod1CKO* mice.

## 3. Discussion

The different aspects of NEUROD1 requirements for pancreas development are not fully understood. Our study revisited the major events of pancreas endocrine development to address the different regulatory roles of NEUROD1. We eliminated *Neurod1* at the onset of pancreas endocrine cell formation and performed molecular and phenotypic analyses of different stages of pancreas development. We show for the first time that the elimination of *Neurod1* significantly affected the transcription factor networks of α and β endocrine lineages during the early phase of the secondary transition of pancreas development, including reduced expression of *Arx, Pdx1, Nkx2.2, MafA, MafB, Insm1, Pax6, Pax4,* and *Pou3f4*. Although the generation of glucagon-producing cells during the first transition was unaffected in the *Neurod1CKO* pancreas compared to littermate controls, most of these cells co-expressed PDX1, indicating abnormalities in the differentiation of α endocrine cells. These results reveal a function of NEUROD1 in early α and β specification and differentiation processes of pancreatic progenitors prior to the endocrine cell expansion stage. Additionally, our study confirms a previous report that the proliferation potential of β cells in the *Neurod1* deletion mutant is reduced, resulting in a diminished β-cell mass at birth [12]. However, we found significant abnormalities in the architecture of islets of Langerhans that cannot be explained only by the defects in the proliferation of β cells. Therefore, we propose that the leading cause of disorganized pancreatic islets and deficiency in β-cell proliferation in the *Neurod1CKO* pancreas is an altered transcriptional network, and thus, that NEUROD1 is required for the early stages of the developmental programs of α and β endocrine lineages, as well as for later stages of endocrine differentiation and maturation.

The role of NEUROD1 as a key transcription factor in the differentiation of β cells was shown in the differentiation of human embryonic stem cells (HESC) from pancreatic progenitors into insulin-producing cells [12]. Similar to our results, the elimination of *Neurod1* in HESCs resulted in a significant reduction of the crucial β-cell transcription factors, including *MafA, Pax6, Nkx2.2, Insm1*, and *Pdx1* [12]. Combinatorial interactions of *Neurod1* and *Nkx2.2* [37], and *Neurod1* with *Insm1* and *Foxa2* are important for β-cell development and function. Furthermore, we showed that the levels of *Arx, MafB,* and *Pou3f4* transcription factors essential for α cell differentiation were reduced together with aberrant co-expression of PDX1 and glucagon in the *Neurod1CKO* developing pancreas, indicating abnormalities in α-cell differentiation. This conclusion is consistent with the role of NEUROD1 in alpha cell lineage specification [37,38]. Surprisingly, we also found a significant reduction of NEUROG3^+^ endocrine progenitors and decreased *Neurog3* expression in the *Neurod1CKO* pancreas during the primary and secondary transitions, at E11.5 and E14.5, respectively. These results suggest a putative interaction between NEUROD1 and NEUROG3 for the coordinated regulation of endocrine cell formation during early pancreas development. Our conclusions are in line with the finding of a study showing that the overlapping and integrative networks of the transcription factors NEUROD1 and NEUROG3 are essential for the differentiation of pancreatic endocrine cells, as the absence or presence of *Neurod1* in the PDX1^+^ pancreatic progenitors affected the lineage potential and altered cell fate decisions [37]. Thus, our results demonstrate an important regulatory function of NEUROD1 for the differentiation of α and β pancreatic endocrine cells prior to the proliferation of these cells. These effects of *Neurod1* deletion on early pancreas development may have previously been overlooked for two reasons: first, because of limited markers [11], and second, due to limited molecular analyses focused only on the later stage of pancreas development, the expansion phase of β cells at E17 [12] and on the formation of β cells [17]. Previously published conditional *Neurod1* deletion mutants used either the BAC transgenic Neurog3-Cre driver [12] with reported Cre activity at E13.5 at the earliest [29,39] or Cre driver under the control of the rat insulin II promoter [40] limiting the deletion of *Neurod1* to developing insulin-producing β cells [17]. Compared to these *Neurod1* mutants, our model of *Neurod1* deficiency in the developing pancreas during the primary transition demonstrates previously unappreciated roles of NEUROD1 in the formation of endocrine precursors and activity of transcriptional networks in developing α and β cells.

A recent study showing that endocrine differentiation and islet morphology are directly related [8] is in line with our conclusion that the abnormalities in the cytoarchitecture of pancreatic islets of *Neurod1CKO* are tied to the changes in the differentiation programs of endocrine α and β cells rather than only as a result of proliferation deficiency. This fact is further supported by the specific deletion of *Neurod1* in developing β cells that did not affect β-cell numbers but disrupted pancreatic islet architecture [17]. Although the mechanisms regulating the formation of the islets of Langerhans during development remain unresolved, a loss of the core–mantle-like cytoarchitecture of islets has been associated with defects in β-cell differentiation [35,41,42,43].

Additionally, we showed that the remaining β cells lacking *Neurod1* have reduced insulin expression resulting from diminished *Ins1* gene transcription. A similar deficiency in *Ins1* transcription was demonstrated in mice with *Neurod1* deletion in developing insulin-producing β cells [17]. Although these mice survive postnatal development and form islets similar in size to those of controls, they are severely glucose intolerant. In contrast to this model, our *Neurod1CKO* mice form disorganized and reduced islets of Langerhans and die of severe diabetes soon after birth. Given the substantial reduction in the proliferation of β cells in the *Neurod1CKO* mutant, we hypothesized that the deficiency in *Ins1* transcription and thus, reduced insulin production, cannot be compensated by increasing the β-cell mass, as reported for the single *Ins1^−/−^* deletion mutant [34], and therefore, contributing to the severe diabetes phenotype of our *Neurod1CKO*.

In summary, our study revisits the central events during pancreas development and provides novel insight into the role of NEUROD1. Our data indicate differential temporal and spatial transcriptional regulatory activities of NEUROD1 during pancreas development. In the course of primary and secondary transitions, *Neurod1* elimination affected the expression of essential transcription factors for the differentiation of α and β lineages, indicating that NEUROD1 is required for the early α and β differentiation similar to human β-cell development. Furthermore, we postulate that changes in the differentiation programs of α and β cells result in a severe diabetic phenotype combining deficiency in insulin expression, diminished proliferation potential of β cells, and disorganized cytoarchitecture of pancreatic islets.

## 4. Materials and Methods

### 4.1. Mouse Model

All experiments using animals were performed according to protocols approved by the Animal Care and Use Ethics Committee of the Institute of Molecular Genetics, Czech Academy of Sciences (protocol code 98/2018 and date of approval 15 January 2019). The experimental mice were housed in a controlled environment (12-h light-12-h dark cycles) with free access to food and water. All experiments were performed with littermates (males and females) cross-bred from two transgenic mouse lines: floxed *Neurod1* (*Neurod1^loxP/loxP^*) [18] and *Isl1^Cre^* (*Isl1*-*Cre*; *Isl1*^tm1(cre)Sev/J^) from The Jackson Laboratory. The expression of Isl1^Cre^ during the formation of dorsal pancreatic bud at E9.5 corresponds to the expression of *Isl1* gene [44], and it is an appropriate Cre driver for *Neurod1^loxP/LoxP^* elimination during the initial stage of endocrine pancreas development. Phenotypes were analyzed on a mixed C57Bl/6 × 129 genetic background and mutants were always compared with littermates. Breeding pairs contain a mouse with two floxed *Neurod1* alleles (*Neurod1*^loxP/loxP^) and a mouse with one floxed *Neurod1* allele together with one *Isl1*^Cre^ allele (*Isl1^Cre/+^;Neurod1*^loxP*/+*^). The *Isl1^Cre^* transgene was transmitted only paternally in order to eliminate any potential influence of maternal ISL1 haploinsufficiency on the developing embryos. Genotyping was performed by PCR on tail DNA. The specific primers used were the following: *Isl1-Cre* F 5′-GCC TGC ATT ACC GGT CGA TGC AAC GA-3′ and *Isl1-Cre* R 5′-GTG GCA GAT GGC GCG GCA ACA CCA TT-3′ with a 700 bp product; *Neurod1* F 5′-ACC ATG CAC TCT GTA CGC ATT-3′ and *Neurod1* R 5′-GAG AAC TGA GAC ACT CAT CTG-3′ with a 400 bp product for the WT allele or 600 bp for the floxed allele. Heterozygous mice *Isl1^Cre/+^;Neurod1^loxP/+^* (*HET*) were comparable to the control mice (*Isl1^+/+^*;*Neurod1^loxP/loxP^* or *Isl1^+/+^*;*Neurod1^loxP/+^*) without any detectable morphological and functional differences. *Neurod1CKO* offspring were recovered at expected Mendelian ratios from E9.5 to birth; chi-square *p* = 0.6131 (117 litters collected and genotyped, 212 *Neurod1CKO*: 241 *HET*: 462 control offspring). Blood glucose levels were measured in animals by glucometer (COUNTOUR TS, Bayer); blood glucose levels maintained above 13.9 mmol/L are classified as diabetic.

### 4.2. Immunohistochemistry

Embryos were dissected in cold PBS, and the dissected pancreas was fixed in 4% paraformaldehyde (PFA) in PBS. For vibratome sections, samples were embedded in 4% agarose and sectioned at 80 μm on a Leica VT1000S vibratome. Samples were then incubated with primary antibodies at 4 °C for 72 h. The primary antibodies used: anti-glucagon (1:400, ab10988, Abcam (Waltham, MA, USA) or 1:500, 4660-1140, BIO-RAD (Hercules, CA, USA)), anti-PDX1 (1:2000, ab47267, Abcam), anti-insulin (1:400, C27C9, Cell Signalling (Danvers, MA, USA) or 1:50, ab7842, Abcam), anti-Ngn3 (1:3, the F25A1B3 monoclonal antibody was obtained from the Developmental Studies Hybridoma Bank, created by the NICHD of the NIH and maintained at The University of Iowa, Department of Biology, Iowa City, IA 52242, USA), anti-alpha-amylase (1:2500, A8273, Sigma-Aldrich (Saint Louis, MO, USA)), anti-Ki67 (1:400, 9129, Cell Signalling), anti-Neurod1 (1:100, sc-1084, Santa Cruz Biotechnology (Dallas, TX, USA)), anti-Cre (1:500, 908001, BioLegend (San Diego, CA, USA)), and anti-peptide C1 and anti-peptide C2 (1:1000 and 1:3000, Beta Cell Biology Consortium). The secondary antibodies used were Alexa Fluor^®^ 488 AffiniPure Goat Anti-Mouse IgG (1:500, #115-545-146, Jackson ImmunoResearch (West Grove, PA, USA)), Alexa Fluor^®^ 594 AffiniPure Goat Anti-Rabbit (1:500, #111-585-144, Jackson ImmunoResearch), and DyLight488-conjugated AffiniPure Mouse Anti-Goat IgG (1:500, #205-485-108, Jackson ImmunoResearch). The nuclei were counterstained with Hoechst 33342. Image acquisition was completed using the Zeiss LSM 880 NLO scanning confocal, with ZEN lite program. The number of insulin^+^, NEUROG3^+^, and Ki67^+^ cells was counted in vibratome sections with the largest pancreatic footprint per photographic field using the Cell Counter plugin of ImageJ program (NIH), as described [45]. We analyzed 6 sections of the pancreas from 3 embryos for each genotype. Tissue paraffin sections (8 μm) of dissected E17.5 and P0 pancreases were treated with 20 μg/mL proteinase K for 20 min at room temperature. The sections were incubated with the TUNEL labeling kit (Roche, Basel, Switzerland) for 1 h at 37 °C, Hoechst 33,342 was used as a nuclear counterstain, and immunolabeled by anti-insulin antibody. The sections were analyzed under a Nikon Eclipse E400 fluorescent microscope.

### 4.3. Reverse Transcription-Quantitative Real-Time Polymerase Chain Reaction

RT-qPCR was performed as described previously [46]. Briefly, total RNA was isolated from the whole pancreas at embryonic day E14.5 or postnatal day P1 (n = 8 samples/group) by Trizol RNA extraction. Following RT, quantitative real-time PCR (qPCR) was performed with the initial AmpliTaq activation at 95 °C for 10 min, followed by 40 cycles at 95 °C for 15 s and 60 °C for 30 s, as described. The *Hprt1* gene was selected as the best reference gene for our analyses from a panel of 12 control genes (TATAA Biocenter AB, Sweden). The relative expression of a target gene was calculated based on qPCR efficiencies and the quantification cycle (Cq) difference (Δ) of an experimental sample (mutant) versus control (the 2^−ΔΔCt^ method). Primers were designed using Primer Blast tool (https://www.ncbi.nlm.nih.gov/tools/primer-blast/ (accessed on 13 July 2018 and 15 January 2018)). Primers were selected according to the following parameters: length between 18 and 24 bases, melting temperature (Tm) between 58° C and 60 °C, G + C content between 40 and 60% (optimal 50%) and efficiency above 80%. Primer sequences are presented in the Appendix A.

### 4.4. Quantification and Statistical Analysis

Data in the figures were represented as mean ± SD (standard error deviation) or ±SEM (standard error of mean). Chi-square test, Student’s *t*-test and One-Way ANOVA were used for statistical comparison between groups that are normally distributed using the built-in function of GraphPad Prism 7 program (GraphPad Software, San Diego, CA, USA). *p*-values of less than 0.05 were considered significant. Significance was determined as *p* < 0.05 (*), *p* < 0.01 (**), *p* < 0.001 (***), or *p* < 0.0001 (****). Sample sizes and individual statistical results for all analyses are provided in the figure legends.

## Figures and Tables

**Figure 1 ijms-22-06713-f001:**
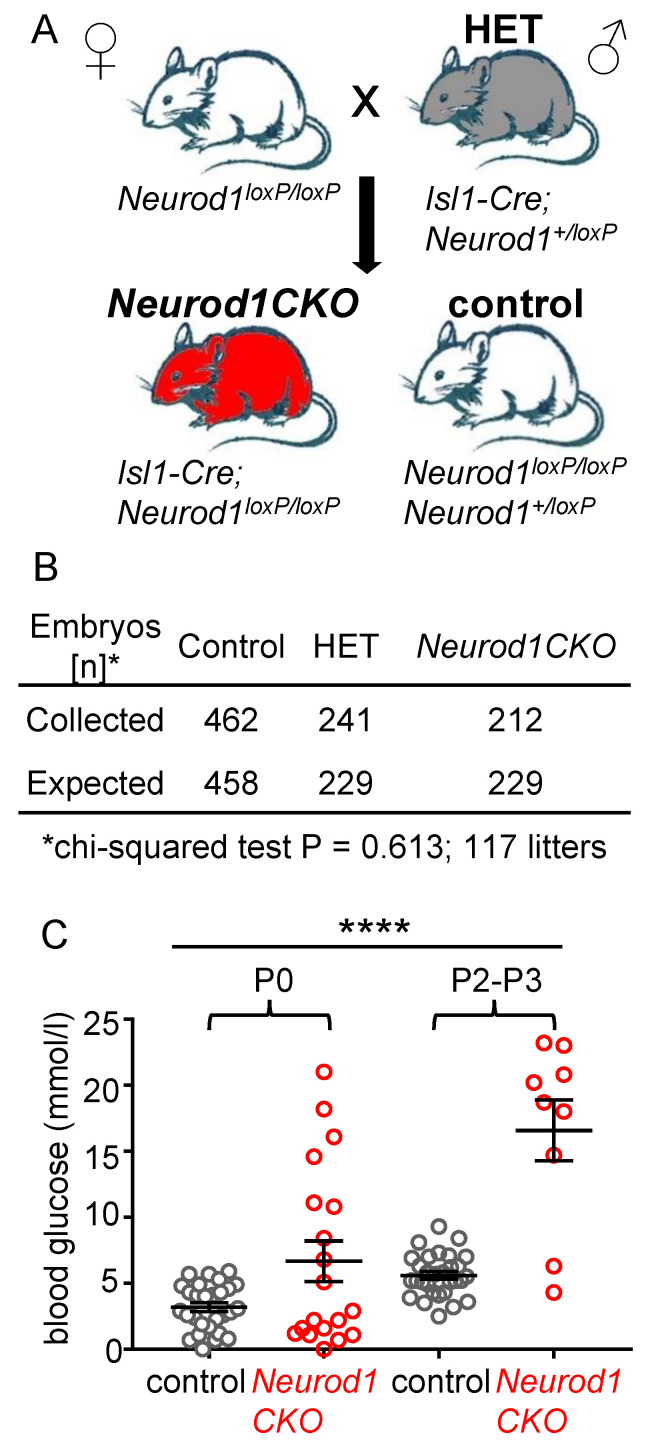
*Neurod1CKO* mutants demonstrate a neonatal diabetic phenotype. (**A**) Breeding scheme shows genotypes for heterozygous (HET), homozygous mutant *Neurod1CKO* and control mice. The *Isl1^Cre^* transgene is transmitted paternally to eliminate any potential maternal influence on the developing embryos. (**B**) The survival of *Neurod1CKO* embryos is not affected, as shown by chi-squared test. (**C**) The blood glucose levels of control and *Neurod1CKO* pups fed *ad libitum* at P0 and P2-P3. Data are presented as mean ± SD, One-Way ANOVA (**** *p* < 0.0001).

**Figure 2 ijms-22-06713-f002:**
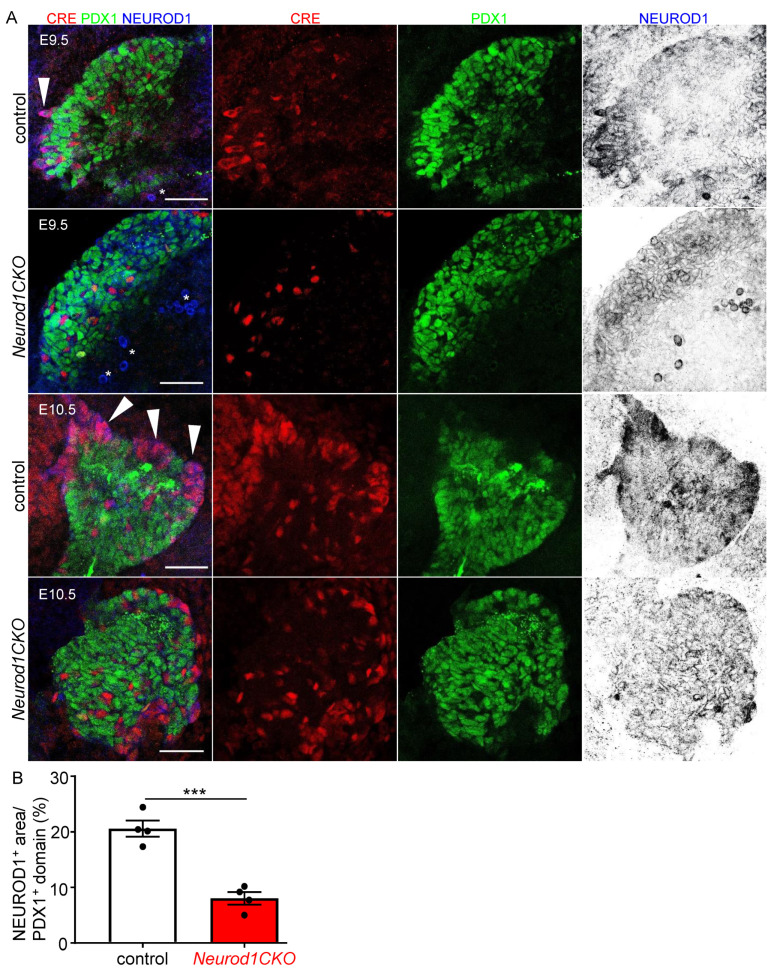
NEUROD1 is efficiently reduced during the primary transition of early pancreas development in *Neurod1CKO*. (**A**) Representative whole-mount immunolabeling of dorsal pancreas shows reduction of NEUROD1^+^ clusters in the PDX1^+^ pancreatic domain (green) of *Neurod1CKO* compare to the control pancreas at E9.5 and E10.5. *Isl1^Cre^* expression is indicated by anti-Cre antibody (red). Large NEUROD1^+^ and Isl1-Cre^+^ cell clusters are detected in the control dorsal pancreas (arrowheads) but not in *Neurod1CKO*. The inverted single-channel image shows NEUROD1 expression. Asterisks indicate autofluorescent red blood cells. Scale bar: 50 μm. (**B**) Quantification of the NEUROD1^+^ area as a percentage of the total PDX1^+^ area in the dorsal pancreas at E10.5 using ImageJ program. Data are presented as mean ± SD, n = 4 embryos/genotype, Unpaired *t*-test (*** *p* = 0.0005).

**Figure 3 ijms-22-06713-f003:**
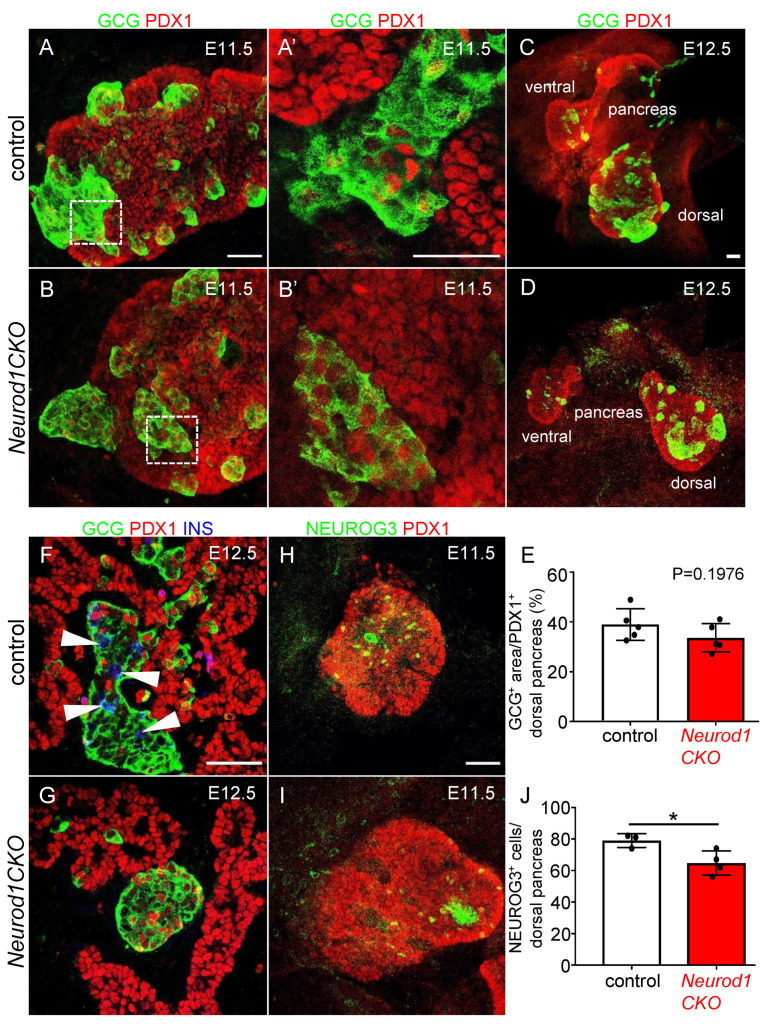
The formation of glucagon^+^ clusters is unaffected but glucagon^+^ cells co-express PDX1 in *Neurod1CKO* during the primary transition. (**A**,**B**) Representative immunolabeling of the first endocrine cells expressing glucagon (GCG) is shown in the dorsal pancreatic epithelium delineated by the expression of PDX1 (whole-mounts). (**A**′,**B**′) Higher-magnification images show increased co-expression of PDX1 and GCG in *Neurod1CKO* compared to control pancreatic epithelium. (**C**–**E**) Representative whole-mount immunolabeling shows dorsal and ventral pancreatic buds with GCG^+^ cells at E12.5. GCG^+^ area in the dorsal pancreas was quantified using ImageJ program and depicted as a percentage of GCG^+^ of the PDX1^+^ area in control and *Neurod1CKO* littermates (**E**). (**F**,**G**) Insulin (INS) expression in endocrine clusters is immunodetected in the control pancreas (arrowheads) but not in *Neurod1CKO* at E12.5 (vibratome sections). (**H**,**I**) NEUROG3 expressing endocrine precursors are shown in the whole-mount of control and *Neurod1CKO* PDX1^+^ dorsal pancreas. (**J**) NEUROG3^+^ cells were counted in the PDX1^+^ dorsal pancreas at E11.5. Data are presented as mean ± SD, Unpaired *t*-test (* *p* = 0.0354). Scale bars: 50 μm.

**Figure 4 ijms-22-06713-f004:**
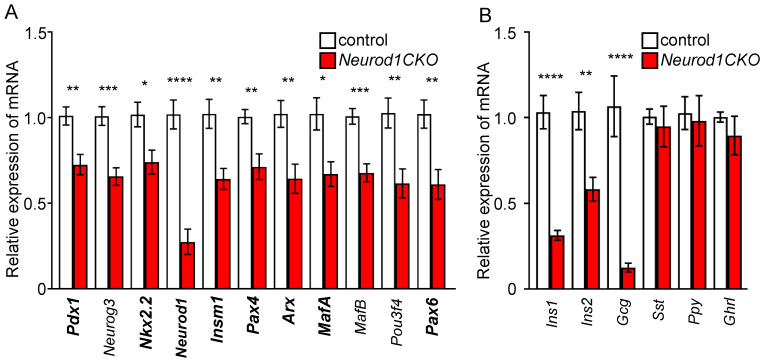
Reduced expression of genes important for the differentiation and function α and β cells in the *Neurod1CKO* pancreas at E14.5. Quantitative RT-PCR analyses of mRNA levels of transcription factors (**A**) and endocrine hormones (**B**) in E14.5 total pancreas of *Neurod1CKO* and control embryos. Genes highlighted in bold print indicate NEUROD1 target genes, as determined by ChIP-Seq analysis of NEUROD1 binding at mouse islet enhancers [31]. Data are presented as mean ± SEM (n = 8), Unpaired *t*-test (**** *p <* 0.0001, *** *p <* 0.001, ** *p <* 0.01, * *p <* 0.05). *Ins1*, insulin 1; *Ins2*, insulin 2; *Gcg*, glucagon; *Sst*, somatostatin; *Ppy*, pancreatic polypeptide; *Ghrl*, ghrelin.

**Figure 5 ijms-22-06713-f005:**
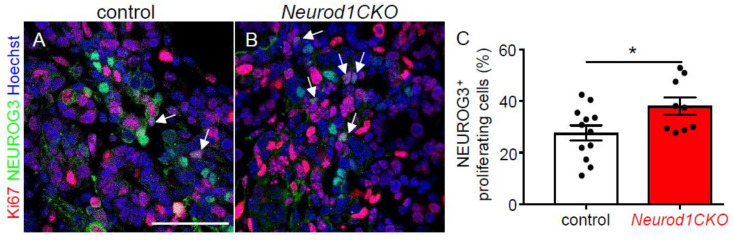
Percentage of proliferating NEUROG3^+^ endocrine precursors is higher in the *Neurod1CKO* pancreas compared to the littermate controls. (**A**,**B**) Immunostaining for proliferating cell nuclear antigen Ki67 (red) in vibratome sections of the control and *Neurod1CKO* pancreas at E15.5 (arrows indicate NEUROG3^+^ proliferating cells). Nuclei are stained with Hoechst. (**C**) Quantification of proliferating NEUROG3^+^ cells per NEUROG3^+^ population (n = 3 embryos per genotype and 3–4 fields per section). Data are presented as mean ± SD, Unpaired *t*-test (* *p* = 0.03). Scale bar: 50 μm.

**Figure 6 ijms-22-06713-f006:**
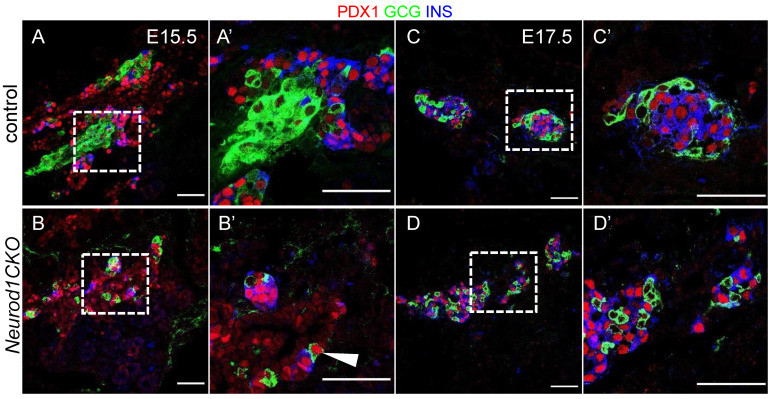
The formation of pancreatic islets is altered in the *Neurod1CKO* pancreas. Representative images of immunolabeling showing glucagon (GCG) producing α cells and β cells expressing PDX1, a marker of differentiated β cells, and insulin (INS) in the control and *Neurod1CKO* pancreas at E15.5 (**A**,**B**), and E17.5 (**C**,**D**). Higher-magnification images show a detail of the architecture of the islets of Langerhans at E15.5 (**A**′,**B**′), and E17.5 (**C**′,**D**′). Note decreased INS production, GCG^+^ cells co-expressing PDX1 (arrowhead), and the disrupted β-cell core/α-cell mantle organization of the islets in *Neurod1CKO*. Scale bars: 50 μm.

**Figure 7 ijms-22-06713-f007:**
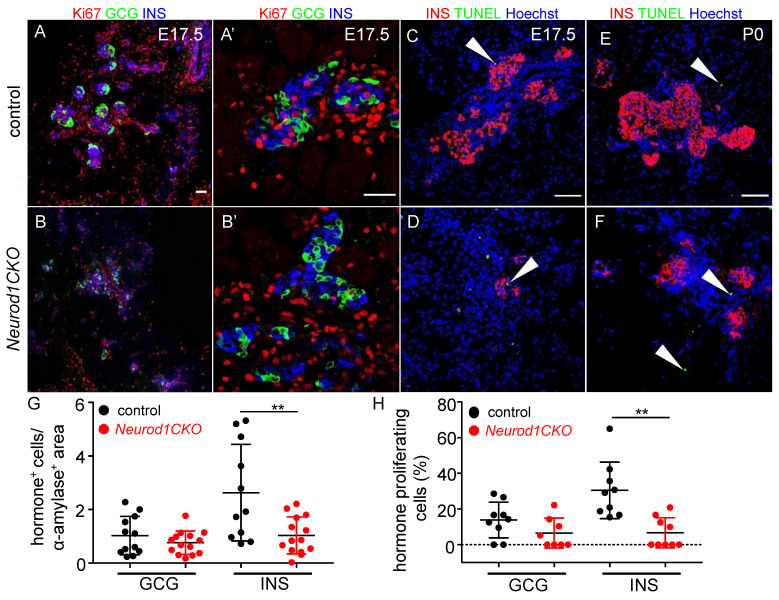
Proliferation potential of β cells is reduced in the *Neurod1CKO* pancreas. (**A**,**B**) Representative sections from the control and *Neurod1CKO* pancreas immunostained for proliferating cell nuclear antigen Ki67 (red) in endocrine α (glucagon, GCG) and β cells (insulin, INS) at E17.5. Note a disrupted formation of the islets of Langerhans in *Neurod1CKO* pancreas (**A**′,**B**′) Higher-magnification images show a detail of proliferating GCG^+^ and INS^+^ cells. (**C**–**F**) Expression analysis of insulin (INS) and TUNEL staining performed on sections of E17.5 and P0 pancreata. Arrowheads indicate TUNEL^+^ cells. Nuclei are stained with Hoechst. (**G**) Relative quantification of GCG^+^ and INS^+^ cells per α-amylase^+^ area (marker of exocrine tissue) and (**H**) the percentage of GCG^+^ and INS^+^ cells expressing Ki67 per total number of GCG^+^ and INS^+^ cells in 80-μm vibratome sections of the pancreas at E17.5. Data are presented as mean ± SD; n = 3/genotype/3–5 fields of view, Unpaired *t*-test (** *p* = 0.0056 (**G**); ** *p =* 0.0011 (**H**)). Scale bars: 50 μm.

**Figure 8 ijms-22-06713-f008:**
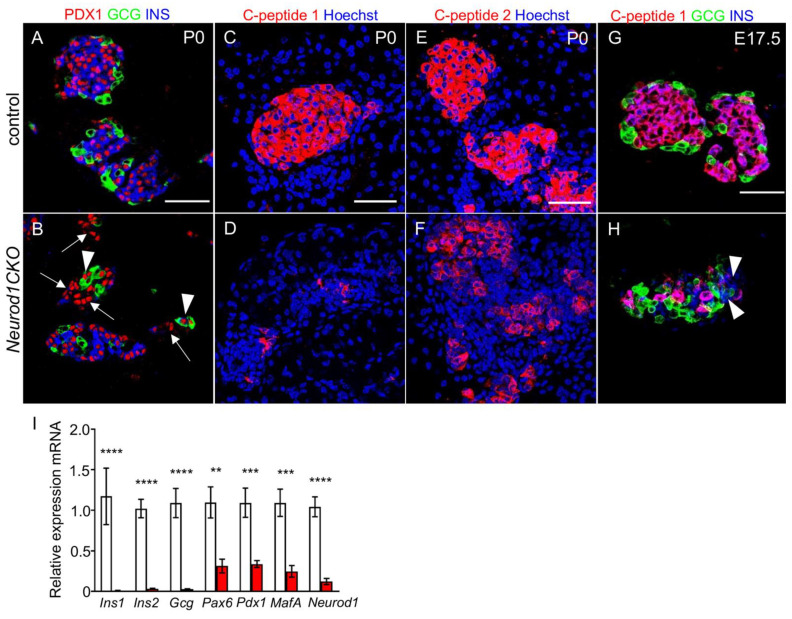
Insulin production is reduced in β cells and the differentiation of α cells is altered in the *Neurod1CKO* pancreas. (**A**,**B**) Representative vibratome sections from the control and *Neurod1CKO* pancreas show the expression of insulin (INS), glucagon (GCG) and PDX1, a marker of differentiated β cells. Note perturbations in the *Neurod1CKO* endocrine α cells co-expressing GCG and PDX1 (arrowheads), and β cells with PDX1 expressing cells without the expression of INS (arrows). (**C**–**F**) Immunostaining for C-peptide 1 and C-peptide 2 shows a significant loss of C-peptide 1 levels in the *Neurod1CKO* pancreas. Nuclei are stained with Hoechst. (**G**,**H**) Reduced expression of C-peptide 1 is noticeable in the *Neurod1CKO* pancreas compared to littermate control at E17.5. Arrowheads indicate INS expressing cells without C-peptide 1 expression. (**I**) Quantitative RT-PCR analyses of mRNA levels of transcription factors and endocrine hormones in P1 total pancreas of *Neurod1CKO* and controls, indicating a major loss of β and α cells in the *Neurod1CKO* mutant. Data are presented as mean ± SEM (n = 8), Unpaired *t*-test (**** *p* < 0.0001, *** *p* < 0.001, ** *p* < 0.01). *Ins1*, insulin 1; *Ins2*, insulin 2; *Gcg*, glucagon. Scale bars: 50 μm.

## Data Availability

Data are contained within the article.

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
