# Peer review of "NEUROD1 Is Required for the Early α and β Endocrine Differentiation in the Pancreas"

_ijms, 2021, doi:10.3390/ijms22136713_

Round 1
Reviewer 1 Report
This is an interesting study demonstrating the effect of conditional Neurod1 deletion on beta and alpha cell expansion. While the study is well cone, there are some points that need to be addressed:
- Please define percent recombination of Neurod1. Neurod1 mRNA is presented, however western blot data demonstrating decreased Neurod1 protein level and histology demonstrating decreased Neurod1+ cells are not presented and need to be. The embryonic time point e14.5 would be best.
- Please show the mRNA level of Nkx6.1 at e14.5. Given its critical role in beta cell development, it is strange that it is overlooked.
- It is unclear the connection of Neurod1 with the genes presented in Figure 4. Are any of these known direct Neurod1 targets? Is the Neurod1 binding site found in the promoter of any of these genes, or are these indirect effects? Please demonstrate using ChIP changes in Neurod1 binding.
- Figure 7 presents the proliferation and apoptosis data for alpha and beta cells. Please present the GCG+ graphs equivalent for G and H, as well as graphs for tunel data.
- Finally, the authors mention that this is not the firs Neurod1 KO animal, nor is it the first Neurod1 conditional knock out. Please discuss 1) Why this version of the KO is necessary, and 2) the differences between this version and the previously published version.s
Reviewer 2 Report
To Authors:
Dear Authors,
The article “ NEUROD1 is required for the early α and β endocrine differentiation in the pancreas ” by R. Bohuslavova et al., aimed to characterize the role of Neurod1 product, the one of the basic helix-loop-helix (bHLH) transcription factors in differentiation of pancreatic islets, particularly of insulin-producing Beta-cells as well as alpha-cells that is known source of insulin-antagonizing peptide - glucagon. The authors demonstrate that the conditional knockout of Neurod1 impairs the expression of key transcription factors for α- and β-cell differentiation, β-cell proliferation, insulin production, and islets of Langerhans formation. The work performed in this paper used experiments were performed with littermates (males and females) cross-bred from two transgenic mouse lines: floxed Neurod1 (Neurod1loxP/loxP)and Isl1Cre (Isl1-cre; Isl1tm1(cre)Sev/J) . Experiments and assays used in the current study are up-to-date and well established. The study is limited by in vivo models and is supplemented by Immunohistochemistry and qPCR assays that add significant value to this work.Nevertheless, there are some question and comments regarding currents work:
Question 1. I would suggest specifying the findings of current study highlighted in lines 22-27 in the introduction part. Because the fact that NeuroD-null mice die of severe diabetes shortly after birth and defects are seen in α and β cells differentiation, that in its turn causes defects in islet formation and loss of majority of β cells have been shown by Naya et al., 1997. Phenotype by conditional knockdown in differentiated cells was published by Gu, C. et al 2010 (Ref #17 in the list). It would be nice to add a paragraph, describing the importance of current work in general view (for example list of markers that are new or differentiation stage where gene silencing takes place).
Question 2. Figure 2A. It is unclear that expression of CRE is activated in PDX1-positive cells. According to representative images in Figure 2 A high expression of CRE corresponds mostly to PDX1-negative cells. Somehow the merged image E9.5 NeuroD1CKO has much better NeuroD1 signal than in isolated blue( from NeuroD1). It seems like the signal from the NeuroD1 channel was artificially reduced, while in merged images we observe different intensity/expression levels.
Question 3. Figure 3A and A'. Please correct me if I have a misunderstanding. Figure 3B' represents the highlighted (zoom in) area of IF image in Figure 3B, whereas in Figure A' IF image does not match with the highlighted area in Figure 3A.
Question 4. Figure 3F/3G demonstrates absence of Ins producing cells on images obtained from Neurod1CKO animals. It would be nice to see if pancreatic KATP channel subunits (Sur1) if it expressed or other markers (ideally NPY) that could provide quantitative values of label progenitor cells, because according to Gu, C. et al 2010 for example Sur1 does not dependent on Neurod1 expression.
Round 2
Reviewer 1 Report
The authors have addressed my previous critiques.
Author Response
Thank you.
Reviewer 2 Report
Dear Authors,
I'm writing to let you know that the most of the questions were addressed and appropriate change were made.
There are few comments regarding text and design issues:
line 33. beta symbol
Figure 1. I would recommend changing color of control events on turkey plot, because bar representing means and SD values are barely seen. In legend 4-star symbol (p<0.0001) is not present of figure.
Figure 2. I would recommend adding scale bars on all merged images.
Figure 6. I would recommend adding scale bars on C/D images.
Figure 7A. Scale bar.
Author Response
Thank you for valuable suggestions. We made all suggested changes in our manuscript.
